# Micro-biopsy for detection of gene expression changes in ischemic swine myocardium: A pilot study

Arvin Chireh[1,☯], Rikard Grankvist[1,☯], Mikael Sandell[1,2,3], Abdul Kadir Mukarram[4], Fabian Arnberg[1,5], Johan Lundberg[1,5], Carsten O. Daub[4], Staffan Holmin[1,5]*

**1** Department of Clinical Neuroscience, Karolinska Institutet, Stockholm, Sweden, **2** Division of Micro and Nanosystems, Royal Institute of Technology, Stockholm, Sweden, **3** MedTechLabs, Solna, Sweden, **4** Department of Biosciences and Nutrition, Karolinska Institutet, Stockholm, Sweden, **5** Department of Neuroradiology, Karolinska University Hospital, Stockholm, Sweden

☯ These authors contributed equally to this work.
* staffan.holmin@ki.se

**Data Availability Statement:** The gene expression data have been deposited in NCBI's Gene Expression Omnibus, and are accessible through GEO Series accession number GSE164962 (https://

## Abstract

Micro-endomyocardial biopsy (micro-EMB) is a novel catheter-based biopsy technique, aiming to increase flexibility and safety compared to conventional EMB. The technique was developed and evaluated in healthy swine. Therefore, the ability to detect disease related tissue changes could not be evaluated. The aim of the present pilot study was to investigate the ability to detect disease related gene expression changes using micro-EMB. Myocardial infarction was induced in three swine by coronary artery balloon occlusion. Micro-EMB samples (n = 164) were collected before, during, and after occlusion. RNA-sequencing was performed on 85 samples, and 53 of these were selected for bioinformatic analysis. A large number of responding genes was detected from the infarcted area (n = 1911). The early responding genes (n = 1268) were mostly related to apoptosis and inflammation. There were fewer responding genes two days after infarction (n = 6), which were related to extracellular matrix changes, and none after 14 days. In contrast to the infarcted area, samples harvested from a non-infarcted myocardial region showed considerably fewer regulated genes (n = 33). Deconvolution analysis, to estimate the proportion of different cell types, revealed a higher proportion of fibroblasts and a reduced proportion of cardiomyocytes two days after occlusion compared to baseline (p < 0.02 and p < 0.01, respectively. S5 File). In conclusion, this pilot study demonstrates the capabilities of micro-EMB to detect local gene expression responses at an early stage after ischemia, but not at later timepoints.

## Introduction

Endomyocardial biopsy (EMB) is an established method for diagnosis of several myocardial diseases, such as rejection after heart transplantation and diagnosis of unexplained heart failure [1]. Despite the value of obtaining cardiac tissue for diagnosis, conventional EMB is limited by a relatively high complication risk (variably reported between less than 1 to 8.9%) and a

www.ncbi.nlm.nih.gov/geo/query/acc.cgi?acc=GSE164962).

**Funding:** This study was funded by Karolinska Institutet (https://www.torstensoderbergsstiftelse.se/), the Stockholm Regional Council (https://www.sll.se), the Söderberg Foundation (https://ragnar.soderbergs.org/), the Erling-Persson Family Foundation (https://familjenerlingperssonsstiftelse.se/en/), all SH. JL was funded by MedTechLabs (https://www.medtechlabs.se/). The funders had no role in study design, data collection and analysis, decision to publish, or preparation of the manuscript.

**Competing interests:** AC, RG, MS, and SH are board members of Microcardix AB, which holds pending patents for the micro-EMB device (Patent Cooperation Treaty application no. WO2020089422A1). RG, JL, and SH own shares in SmartCella Holding AB, whose subsidiaries have interests in catheter technology and regenerative cardiology. This does not alter our adherence to PLOS ONE policies on sharing data and materials.

low diagnostic yield [2–4]. To address these shortcomings, we previously developed a miniaturized and flexible micro-biopsy (micro-EMB) device, aiming to increase safety and navigability to all parts of the heart [5]. We also devised a low-input RNA isolation protocol to enable RNA-sequencing (RNA-seq) analysis on small tissue samples. The technique was evaluated in swine, demonstrating reduced trauma and increased navigability compared to conventional methods, as well as good RNA-seq data quality [5]. However, the previous study was performed in healthy animals and did not assess the ability to detect gene expression changes related to disease.

The aim of the micro-EMB procedure is to provide a safe and accurate tool for molecular diagnosis of disease and research of pathophysiological responses. Therefore, its ability to detect disease related changes is essential. Several studies suggest that gene expression profiling can be used for both research and clinical diagnosis [6–9]. For instance, Halloran and colleagues have described a transcriptomics-based test (MMDx™) to diagnose transplant rejection [8]. This demonstrated the use of molecular tests as potential alternatives to conventional analysis methods such as histology. However, MMDx™ relies on conventional EMB and is therefore limited by the same flexibility and safety issues as discussed above. To our knowledge, the novel micro-EMB technique is the first attempt to miniaturize a biopsy device specifically intended for low-input molecular analyses. Due to the dramatic reduction in tissue specimen size—over three orders of magnitude (from the conventional 10–40 mg to approximately 0.01 mg [5, 10])—it remains unknown whether the tissue samples can be used to detect gene expression changes.

To this end, the present study tests the hypothesis that micro-EMB can be used to detect gene expression changes in a cardiac disease model. For this purpose, we chose a balloon occlusion-reperfusion model of myocardial infarction (MI) in swine. Since the study was not targeting a specific cardiac disease, virtually any cardiac disease model could have been used. Numerous large animal models of myocardial disease have been established [11], and we chose MI due to a relatively straightforward procedure and a rich existing reference literature. Previous studies of gene expression changes after MI have shown early changes in genes related to inflammation and immune responses, as well as later changes related to extracellular matrix (ECM) remodeling [12, 13].

In this pilot study, we show for the first time that the micro-EMB procedure can detect early gene expression changes following MI, consistent with previous findings from RNA-sequencing studies, despite the large reduction of sampled tissue amount.

## Results

A myocardial infarction (MI) swine model was used as disease model. MI was induced for one hour, followed by reperfusion (Fig 1A). During occlusion, the average systolic blood pressure was 80 mmHg (SD 8.0), heart rate 99 bpm (SD 6.8), and oxygen saturation 96% (SD 0.7; S1 File). All three animals tolerated the procedure and were studied up to 14 days according to the study plan. There were no signs of severe complications such as tamponade or sustained arrhythmia during occlusion or micro-EMB procedures, and the animals behaved normally between the procedures. Myocardial injury was confirmed by troponin dynamics and necropsy (Fig 1B, S1 Fig). No significant pericardial effusion or bleeding to the pericardium was noted on necropsy.

Micro-EMB samples from the left ventricle (LV) were repeatedly taken during the experiments (n = 164). Samples were taken at baseline (n = 15), during the one-hour occlusion (n = 14), up to two hours after re-perfusion (n = 49), after two days (n = 33), and finally after 14 days (n = 53; S2 File). Most micro-EMB (65%) were aimed towards the infarcted apical

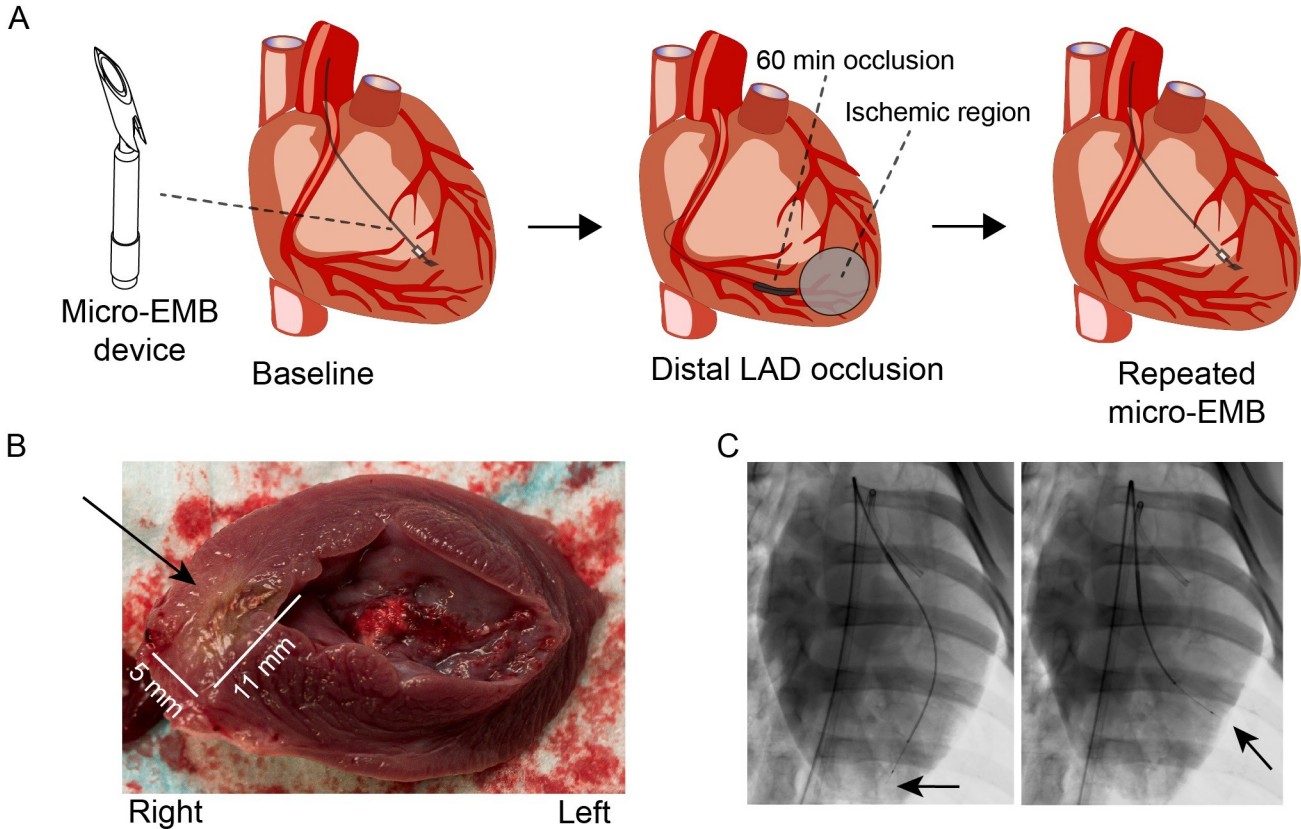

**Fig 1. Experimental design.** (A) Baseline micro-EMB was followed by a 60 minutes occlusion of distal left anterior descending artery. Repeated micro-EMB was performed before, during, and after occlusion at 1–3 hours, two days and 14 days. (B) Post-mortem image of the resulting necrotic area in the right peri-apical wall (approximately 5 x 11 mm). (C) Fluoroscopy in posteroanterior view, showing sampling from apical (infarcted) and lateral (non-infarcted) regions (arrows). Panel A contains elements reproduced from Springer Nature [5] under Creative Commons License CC-BY 4.0.

region, whereas the rest (35%) were aimed towards a basolateral non-ischemic location (Fig 1C, S2 File). Repeated sampling with micro-EMB did not cause any detectable LV wall perforation, clinically significant pericardial effusion, or sustained arrhythmia at two days after infarction, a timepoint with known high risk for arrhythmias [14, 15], nor at day 14. Transient ventricular extra beats, without hemodynamic effect, were observed in all animals during penetration of the endocardium with the biopsy device.

A subset of samples (n = 85), representing all timepoints, was used for RNA-seq (Table 1). Quality control was conducted on the RNA-seq data before downstream analyses. Twenty-four samples were excluded due to failed quality control and two samples were excluded due to blood contamination. Principal component analysis (PCA) showed clear visual separation between heart tissue samples and blood (S2 Fig). The plot revealed a group of outlier samples (n = 6), which turned out to have an up-regulation of collagen related genes (S3 File). This finding indicated that these samples were harvested from fibrous compartments of the heart and were therefore excluded to avoid noise. The remaining 53 samples were included in the downstream analyses.

Differential expression (DE) analysis was performed to explore gene expression changes after occlusion. Initially, only samples harvested from the ischemic region (near apex) were considered (n = 24). The analysis revealed 1911 significantly regulated genes (S3 File). These genes were related to biological processes such as "apoptosis", "immune response" and

**Table 1. Number of samples from each timepoint, before and after exclusion.**

| Sample Type | Timepoint | Sequenced samples (n) | Samples after exclusion (n) |
|---|---|---|---|
| **Micro-EMB (ischemic apex)** | Baseline | 6* | 3* |
| | 0–1 h | 6 | 4 |
| | 1–3 h | 11 | 8 |
| | 2 days | 9 | 3 |
| | 14 days | 9 | 6 |
| **Micro-EMB (non-infarct region)** | Baseline | 6* | 3* |
| | 0–1 h | 3 | 3 |
| | 1–3 h | 6 | 2 |
| | 2 days | 6 | 4 |
| | 14 days | 6 | 6 |
| **Blood** | Baseline | 6 | 1 |
| | 0–1 h | 0 | 0 |
| | 1–3 h | 5 | 3 |
| | 2 days | 6 | 5 |
| | 14 days | 6 | 5 |

* The six baseline samples were not specifically targeted to any region of the heart and were therefore shared between infarcted and non-infarcted samples.

"inflammation" (Fig 2A, S4 File). To understand how these regulated genes were expressed in each sample, we generated a heatmap of the top regulated genes (n = 1000; Fig 2B). The samples clustered into two groups, separating samples from the early phase after occlusion (all 1–3 h samples and one 0–1 h sample) from the rest. All samples taken at 1–3 h from all three swine were represented in this distinct cluster.

For comparison, DE-analysis was also performed on samples from the non-ischemic region of the heart (n = 18, Table 1). A total of 33 genes were significantly regulated between all timepoints (S3 File), which is notably fewer than the 1911 genes in the previous group. The regulated genes were related to processes such as "apoptosis" and "cell death". To understand the relationship between samples from the infarcted and non-infarcted areas, a heatmap was generated, including all samples from both regions of the heart (Fig 2C). The heatmap showed that two samples from the non-infarcted region clustered with the 1–3 hour samples from the infarcted region. Both samples were from the same animal. In other words, two out of 18 non-infarcted samples had similar gene expression patterns as the markedly inflamed and apoptotic samples from the infarcted area.

Next, we identified time-specific changes by comparing gene expression at individual time points with baseline samples. As previously, only samples from the ischemic region were initially considered (n = 24). Our analysis revealed regulated genes between baseline and 0–1 hours, 1–3 hours and two days, respectively (Table 2, S3 File). Regulated genes were related to processes such as "coagulation", "apoptosis", "inflammatory response" and "cell migration". These results further confirmed that a local injury and inflammation was captured from the infarcted region in the first three hours following the ischemic event, but less so at later time points. The analysis was repeated on samples from the non-ischemic region, revealing only a single gene changing expression between baseline and 1–3 hours, but no genes between baseline and any of the other timepoints (Table 2, S3 File).

To visualize the dynamic changes of some of the regulated genes, the top five regulated genes specific to each timepoint were plotted longitudinally (Fig 3). The plots show that the top regulated genes for the first hours after occlusion, such as *MYC*, *ARC* and *PDK4* were

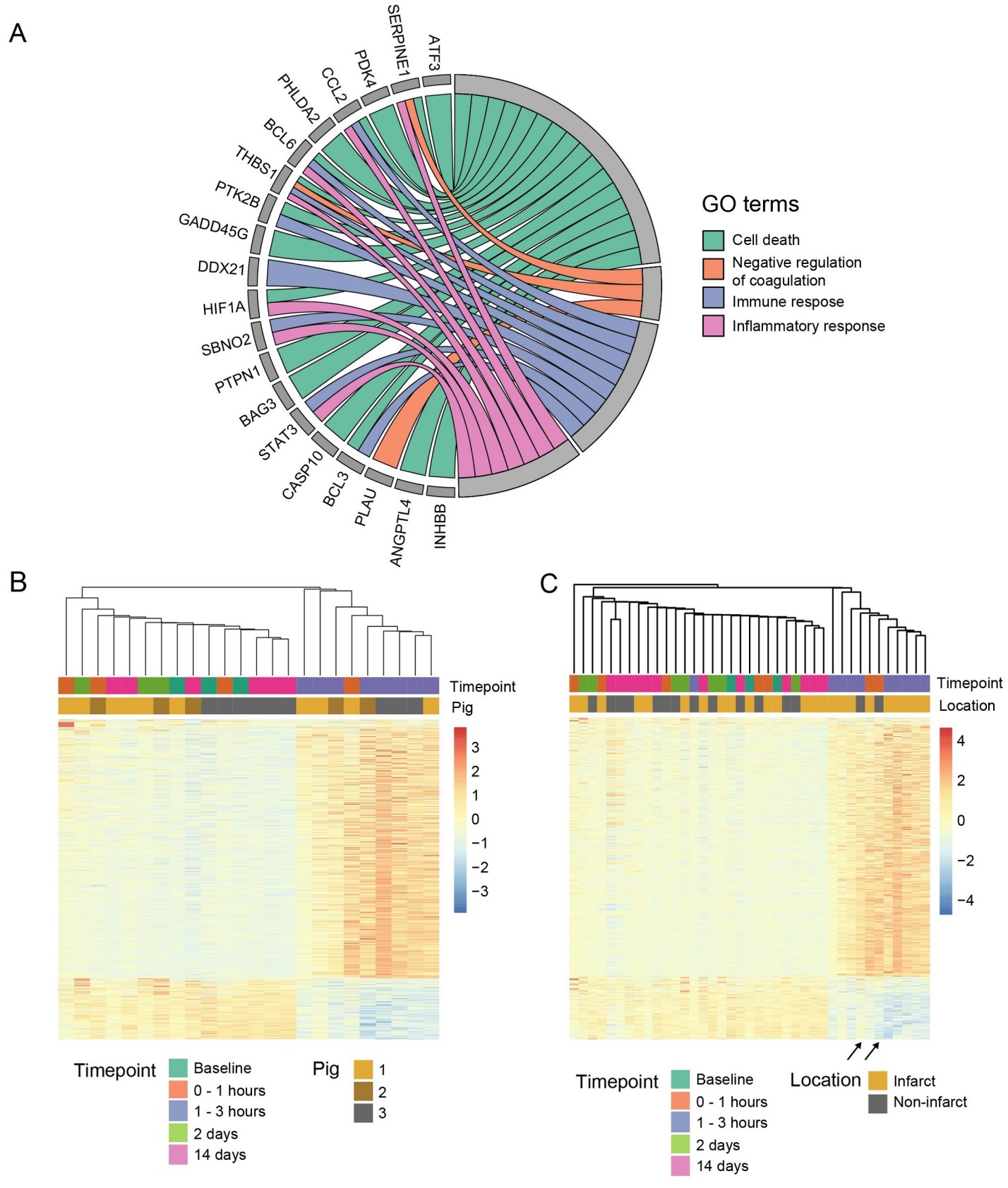

**Fig 2. Differential expression analysis.** (A) Chord plot of top regulated genes and corresponding enriched biological processes from differential expression analysis between infarcted samples (n = 24). (B) Heatmap of top regulated genes (n = 1000) with hierarchical clustering showing two main groups, where the right cluster is exclusively composed by samples from the first hours after the occlusion. Values are normalized expression values centered and scaled in the row direction. (C) Heatmap of the same genes as in B, but also including samples from the non-infarction region (n = 18). Two samples from non-infarction region (arrows) are clustered with infarcted 1–3 hour samples.

markedly up-regulated in both the 0–1 hour and the 1–3 hour group. For the day two genes, *FN1* and *CILP* were specifically up-regulated, whereas *TNC* and *FAM111A* were also upregulated after three hours. These results confirm that the genes were dynamically up- and down regulated with some overlap between adjacent timepoints.

Evidently, the gene expression analyses revealed distinct changes at the early timepoints after the ischemic event, but to a lesser extent at later timepoints. To look further for later changes, we explored the literature for known markers. In a recent gene expression study on swine myocardium, Hinkel and colleagues used a deconvolution strategy to estimate the proportion of different cell types in swine myocardium after an infarction [13]. They found increased proportions of macrophages and fibroblasts in myocardial samples 33 days after a large infarction, compared to non-infarcted controls. To test whether this finding was present in our data, as an indication of late tissue changes, we used the same deconvolution strategy (Fig 4, S5 File). The results showed an increased estimated proportion of fibroblasts and a reduced proportion of cardiomyocytes after two days compared to baseline (unpaired t-test, adjusted p value < 0.02 and < 0.01 respectively). The same difference was found between day two and 14 (unpaired t-test, adjusted p value < 0.02 and < 0.01 respectively), but not between baseline and day 14 (p > 0.05). There was no significant difference for macrophages or endothelial cells (p > 0.05). Only samples from the ischemic region at baseline, day two and day 14 (n = 12) were considered for this analysis.

## Discussion

The recently described myocardial micro-biopsy (micro-EMB) procedure is the first attempt to miniaturize conventional EMB in order to increase flexibility and safety, while still generating robust and useful molecular data from all parts of the heart. Coupled with RNA-seq, micro-EMB can potentially become a useful asset in clinical diagnosis and research investigations. However, a key question towards its usability was whether it can be used to detect disease related gene expression changes, despite the small sample sizes. The results from this pilot study clearly suggest that it indeed can, as early inflammatory changes after an ischemic event were distinctly identified. All samples harvested from the infarcted region after 1–3 hours clustered together and showed a clear up- or downregulation of a specific set of genes. These samples represented all three swine, demonstrating reproducibility.

**Table 2. Significantly regulated genes and enriched biological processes at different timepoints.**

| Comparison | Sampling location | Number of regulated genes (up, down) | Examples of enriched processes |
|---|---|---|---|
| **0–1 h vs baseline** | Infarcted region | 28 (28, 0) | Coagulation, apoptosis |
| | Non-infarcted region | 0 | - |
| **1–3 h vs baseline** | Infarcted region | 1240 (881, 359) | Cell death, immune response, inflammatory response, coagulation. |
| | Non-infarcted region | 1 (1, 0) (*ZBTB16*) | T-cell differentiation |
| **2 days vs baseline** | Infarcted region | 6 (6, 0) | Cell migration, calcium-independent cell-matrix adhesion |
| | Non-infarcted region | 0 | - |
| **14 days vs baseline** | Infarcted region | 0 | - |
| | Non-infarcted region | 0 | - |

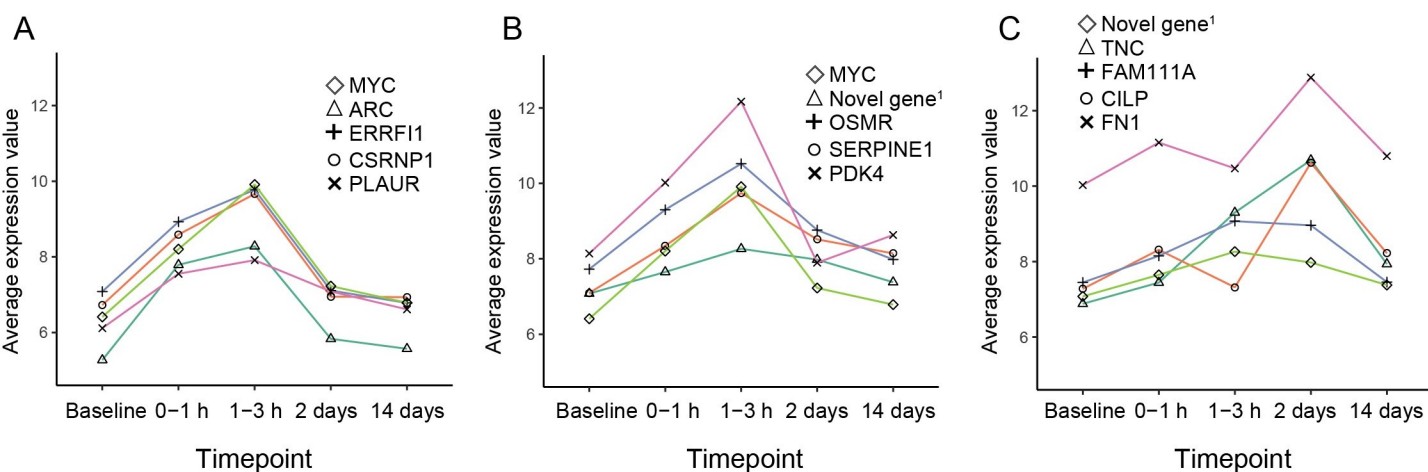

**Fig 3. Longitudinal plots of average gene expression values (normalized and transformed counts) for top regulated genes from each timepoint compared to baseline.** (A) Top genes (n = 5) for the 0–1 hours timepoint. (B) Top genes (n = 5) for the 1–3 hours timepoint. (C) Top genes (n = 5) for the two days timepoint. Novel gene[1] = *ENSSSCG00000014997*.

Early responding genes were related to processes such as apoptosis and inflammation, which is consistent with previously described findings [16, 17]. The capability to detect disease-related changes was highlighted by the fact that all samples harvested at 1–3 hours showed similar expression patterns. However, there were fewer regulated genes at day two and none at day 14. In a recent study of myocardial infarction in swine, significant differences were detected as late as 33 days after a large infarction [13]. However, the infarction created in that specific study was significantly larger and also caused heart failure. The relatively small infarction used in this study, few sample replicates, and the limited visual guidance provided by conventional fluoroscopy alone, could have contributed to difficulty in detecting later changes. Deconvolution analysis suggested increased presence of fibroblasts and conversely a reduced proportion of cardiomyocytes after two days compared to baseline. These findings suggest that there is a capability to detect relatively late changes despite the small infarction size. However, since the estimation of cell proportions is still based on the gene expression profiles, this data suffers from the same limitations as the gene expression analysis. Since conventional EMB was not used in the current study, it cannot be concluded whether the failure to detect later changes was influenced by the reduction of sample tissue size, compared to larger samples from the conventional device.

Analysis of samples from the non-infarcted myocardial regions revealed only 33 regulated genes. This was expected considering the small infarction. Surprisingly, two distant samples clustered with 1–3-hour samples taken from the infarcted region (Fig 2C). This finding most likely represents sampling inaccuracy. It seems that two of the distant samples were in fact harvested from or near the ischemic/infarcted area. As mentioned, when using basic fluoroscopic guidance, the ability to detect and target an ischemic lesion is limited. Sampling accuracy may improve with the use of other guidance techniques, such as cardiac magnetic resonance (CMR) imaging or an electroanatomical mapping system [18–20]. We did not have access to these techniques when the study was performed.

This study also highlights the capability to safely study individuals (animals or human) longitudinally with micro-EMB. Due to the reduced trauma, the method allows repeated sampling over time with low risks of complications. This is supported by the fact that all three swine in this study survived and did not suffer from any major complication such as tamponade or sustained arrhythmia, despite extensive sampling during and soon after occlusion of the left

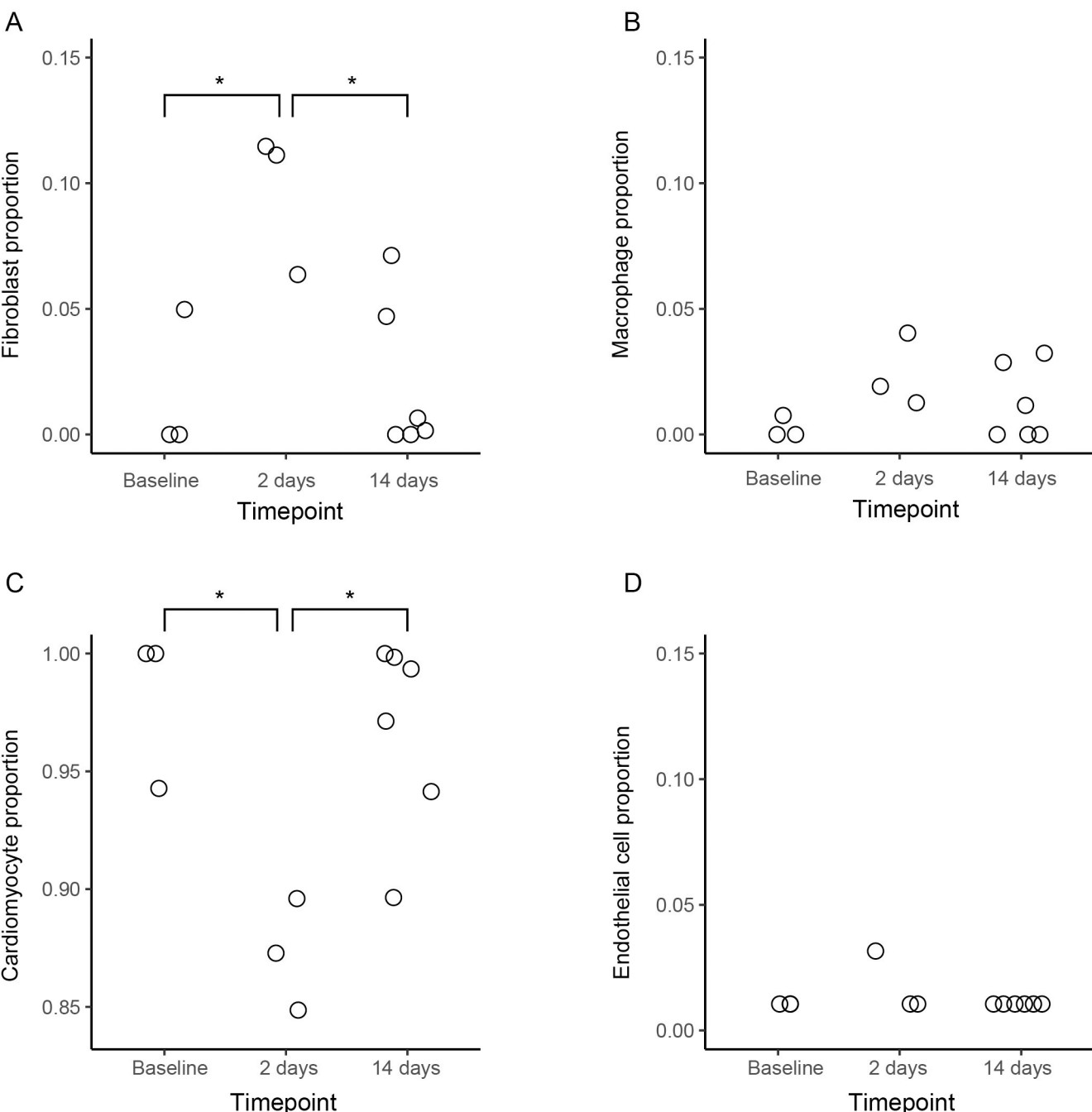

**Fig 4. Estimated proportions of cell types in myocardial samples (n = 12).** (A) Estimated proportion of fibroblasts. Significant difference between baseline and day two samples (unpaired t-test, adjusted p value < 0.02), as well as between day two and day 14 samples (adjusted p value < 0.02). (B). Estimated macrophage proportion. No significant differences between groups (adjusted p values > 0.05). (C) Estimated proportion of cardiomyocytes. Significant difference between baseline and day two samples (adjusted p value < 0.01), as well as between day two and day 14 samples (adjusted p value < 0.01). (D) Estimated endothelial cell proportion. No significant differences between groups (adjusted p values > 0.05).

anterior descending artery (LAD). This finding is noteworthy given the known myocardial sensitivity in swine [14, 15]. Our study also highlights the flexible navigation possibilities with the new miniaturized device. Using pre-shaped guiding catheters (4–5 French), or low-profile deflectable guiding catheters, allows sampling from different parts of the ventricular walls.

Whether these capabilities translate to the clinical situation in humans with chronic heart disease must be studied in larger animal studies and clinical trials before conclusions can be drawn.

The study was limited by relatively few replicates from each timepoint. It is known from our previous work that the inclusion rate of sequenced micro-EMB procedure is around 80%, due to small and extremely selective sampling, which has to be accounted for when preparing analyses [5]. The study was also limited by partly inaccurate sampling of infarcted and non-infarcted areas, which could potentially be addressed with supplemental endovascular guidance techniques. As this was a pilot study and not intended to develop a diagnostic test, relatively few animals were enrolled, and there was no control group. Additional replicates would have been required for development and cross-validation of a diagnostic test. Moreover, the rate and number of biopsies was not adapted to a clinical context, as the sampling was more extensive than what is typically performed in the clinic. It should also be noted that transcriptomic analysis is not the only method for molecular tissue analysis. For instance, previous studies have demonstrated proteomic changes after myocardial infarction [21]. However, proteomic analyses generally require larger tissue samples and are also currently less explored in a clinical context compared to transcriptomics. Due to the small size of micro-EMB tissue samples (around 10 μg), proteomic assays are currently challenging to implement.

In conclusion, this pilot study showed that micro-EMB with RNA-seq can capture early gene expression changes after cardiac ischemia, demonstrating its potential for future clinical and research applications. Future studies could focus on improving the sampling guidance and test the diagnostic performance in relevant diseases, compared to other methods.

## Materials and methods

### Study design

The purpose of this pilot study was to evaluate the ability to capture gene expression changes in a myocardial disease model, using a micro-EMB device (Fig 1A). Three female mixed breed Yorkshire-Swedish farm pigs (weight 34.2–36.5 kg) were used. Baseline micro-EMB samples were acquired before a balloon catheter was placed in the distal LAD. The balloon was inflated for 60 minutes before re-perfusion. The occlusion time was selected to cause infarction in at least some of the tissue, while simulating a reasonable clinical scenario with reperfusion treatment. Continuous arterial blood pressure, ECG and oxygen saturation was measured before, during, and after occlusion. From the point of inflation and onwards, micro-EMB samples were acquired continuously up to three hours from inflation (approximately every 3–5 minutes). Some biopsies were harvested apically in the left ventricle (LV), from the presumed ischemic or infarcted region, whereas others were harvested more laterally and basal from a non-ischemic region. Additional micro-EMB samples from both regions were acquired at two days and 14 days after the initial intervention. The number of harvested biopsies was selected to maximize statistical power.

Myocardial infarction was confirmed by ECG changes, visual inspection of the myocardium post-mortem, and by troponin assays. Histology was not considered necessary, as the presence of infarction was apparent on necropsy.

A subset of the harvested micro-EMB samples was selected for RNA-seq. The RNA-seq data was analyzed to detect relevant gene expression changes at different timepoints.

### Ethical considerations and animal husbandry

All research was conducted in accordance with national and local guidelines for Sweden and Karolinska Institutet respectively. All animal experiments had prior ethical approval from the local ethics committee (Stockholms Norra Djurförsöksetiska Nämnd, Stockholm, Sweden). All

experiments were performed in accordance with 'Principles of Laboratory Animal Care' formulated by the National Society for Medical Research as well as the 'Guide for the Care and Use of Laboratory Animals' prepared by the Institute of Laboratory Animal Resources.

## Intervention and monitoring of complications

Animals were pre-medicated with sedatives (intramuscular injection of tiletamine 2.5 mg/kg, zolazepam 2.5 mg/kg and medetomidine 0.1 mg/kg) and taken to an angiography suite (Allura XD20, Philips, Amsterdam, the Netherlands). The animals were intubated and mechanically ventilated while receiving standard surgical anesthetic care. Anesthesia was induced with i.v. propofol (20 mg) or i.v. sodium pentobarbital (120–180 mg) and maintained with sodium pentobarbital infusion (15–20 mg/kg/hour) or inhaled isoflurane (0.5–1.5%). Analgesia was achieved using i.v. fentanyl (100 µg/hour). Throughout intervention, the animals were monitored for complications using three-lead ECG, continuous arterial blood pressure measurement, oxygen saturation, temperature, and urine production. Any complications during intervention were noted, such as abnormal blood pressure, oxygen saturation, and sustained arrhythmia (defined as non-sinus rhythm for over 30 seconds and/or requiring treatment). Animals were treated with 75 mg amiodarone i.v. twice: just before intervention and after one hour. Venous blood samples were acquired before and after ischemia to evaluate myocardial troponin. After intervention, animals were treated with buprenorphine 0.01–0.05 mg/kg i.m. and meloxicam 2–3 mg/kg i.m, once daily as analgesia. Between biopsy sessions, animals were monitored by laboratory animal care staff specifically trained in large animal husbandry, under veterinarian supervision. Any symptoms or complications were noted.

At the initial intervention, baseline micro-EMB samples were first acquired, followed by coronary angiography using a 6 F Convey (Boston Scientific, Marlborough, MA, USA) hockey stick guide catheter to verify normal coronary artery anatomy. A microwire (Choice floppy, Boston Scientific) was placed in the LAD and a 15 mm x 1.5–2 mm balloon catheter (Emerge, Boston Scientific) was advanced just beyond the third diagonal artery. The balloon was inflated, and a left coronary artery angiography was performed to confirm balloon occlusion. After 60 minutes, the balloon was deflated, and a left coronary artery angiography was repeated to verify reperfusion.

All biopsies were obtained using a transfemoral approach, using the same technique as described previously [5]. In short, a 5 F or 7 F short sheath (Saint Jude Medical, Little Canada, MN, USA) was used to gain access to the femoral artery, and a standard 1.66 mm (5 F) or 1.33 mm (4 F) straight diagnostic catheter (Torcon NB advantage, Cook Medical, Bloomington, IN, USA) was used to access the LV. The micro-biopsy samples were taken by advancing a 2.7 F micro-catheter housing the biopsy device into the LV. In the LV, the micro-biopsy device alone was advanced into the myocardium. Inside the myocardium, the cutting tip of the device was advanced, rotated 180 degrees, and retracted to pinch off a tissue sample. After retraction of the biopsy device, the samples were retrieved from the device, with aid of a surgical microscope. The samples were stored in 0.2 ml PCR tubes and immediately placed on dry ice before storage at -80˚C.

For follow-up biopsies on day two and 14, animals were pre-medicated and anesthetized as above. Biopsies were taken in the same manner, targeting both the apical, previously infarcted region, and the basolateral non-infarcted myocardium. After biopsies on day two and 14, left ventriculography was performed to evaluate tamponade, gross left ventricular dysfunction, and ventricle wall rupture. After the final biopsy at day 14, the animals were sacrificed by overdose of i.v. sodium pentobarbital (100 mg/kg), and the heart was examined grossly for pathology. Post-mortem examination included assessment of pericardial effusion, ventricular wall

injury, infarction, valve damage, intraventricular thrombus, and mural hematoma. Significant pericardial effusion was defined as >20 ml of fluid in the pericardium, and any blood in pericardium was noted.

## Sample preparation and RNA-sequencing

RNA was isolated using the same technique as previously described [5]. For total RNA isolation, a modified version of Agencourt RNAdvance tissue kit (Beckman Coulter, Indianapolis, IN, USA) was used. The kit protocol was modified by reducing all reagent volumes to 1/13. In the homogenization step, the samples were vortexed and centrifuged with standard benchtop equipment. Any remaining bulk tissue, after homogenization, was discarded before proceeding with the protocol. The RNA isolation was performed in different batches, to minimize bench-time for each sample. The samples were randomly assigned to each batch, using a random number generator in Microsoft Excel. Libraries were prepared with the SMARTer Pico Stranded Total RNA-Seq Kit (Takara Bio Inc.). Libraries were pooled and sequenced on the Illumina NovaSeq platform (Illumina, San Diego, CA, USA) using 2×50 base pair reads. After generation of gene expression data, PCA plots were colored and shaped based on batch number to confirm that the expression patterns were not driven by batch effects.

## Data analysis and statistics

Raw sequencing reads were mapped to the pig genome assembly Sscrofa11.1, using STAR RNA-seq aligner [22]. The package featureCounts [23] was used to assign the reads to genes. All downstream analyses, starting from the gene count matrix, were performed using the R statistical computing language, version 4.0.2. For all visualizations, gene counts were transformed and normalized with DESeq2, using variance stabilizing transformation [24].

Mitochondrial, ribosomal, and lowly expressed genes (defined as count per million less than 1 in at least 20% of the samples) were excluded, leaving 22843 genes in the count matrix. Samples with abnormal quality control parameters, such as total number of mapped reads or percentage of genes mapped to features, were excluded. Blood contaminated samples were identified with a panel of blood characteristic genes, using a similar method as previously described [5]. Outlier samples with regard to gene expression patterns (n = 6) were observed in the initial PCA and characterized by differential expression (DE) analysis. Analysis revealed up-regulation of genes related to collagen, indicating high fibrous content compared to the remaining myocardial samples. These samples were excluded. PCA was used to confirm the data quality of the remaining samples and to look for batch effects, which were not present.

DE analysis was performed using the DESeq2 package [24]. Myocardial samples from the ischemic region were compared with a global model, using timepoint as the only term in the design formula. A likelihood ratio test was performed with default parameters. Specific timepoints were compared by individual Wald tests. Samples from the non-ischemic regions were analyzed separately, using the same methods and thresholds as for the infarcted samples.

Samples were clustered by unsupervised hierarchical clustering. Enrichment analysis was performed with the g:Profiler web tool [25], using the Benjamin-Hochberg (BH) method for multiple test correction. To estimate the proportion of different cell types in the myocardial samples, a deconvolution technique was used, using the CIBERSORTx tool [26]. A signature matrix was provided by Hinkel and colleagues, based on bulk and single-cell RNA-sequencing data from mouse [13]. The signature matrix was uploaded together with our gene expression matrix in transcripts per million (TPM). The resulting table of estimated cell type proportion was used for plotting and statistical analysis with t-tests. Correction for multiple testing was done with the BH method. Adjusted p-values of 0.05 or less were considered significant.

## Supporting information

**S1 Fig. Troponin-T assays.** Troponin-T assays from venous blood drawn at different time-points show a distinct surge in all swine (n = 3) at three hours after left anterior descending artery occlusion, indicating myocardial injury.
(TIF)

**S2 Fig. Distribution of samples including outliers.** (A) PCA plot of top 500 high variance genes on all samples after excluding technically failed samples (n = 59). The plot shows good separation between blood and myocardial samples, apart from outlier samples marked by a dashed rectangle (n = 6). (B) Heatmap of normalized expression of genes up-regulated in heart samples (n = 198, top rows) and the outlier samples (n = 206, bottom rows). All heart samples were included (n = 45). Outlier samples indicated in (A) constitute a separate cluster (bottom bracket).
(TIF)

**S1 File. Hemodynamic parameters measured during LAD occlusion.**
(XLSX)

**S2 File. Number and characteristics of all biopsies.**
(XLSX)

**S3 File. Lists of outlier and DE genes at different timepoints and locations.**
(XLSX)

**S4 File. Output from gene ontology analysis.**
(CSV)

**S5 File. Deconvolution data.**
(XLSX)

## Acknowledgments

We thank Pellina Jansson for technical assistance during the animal procedures. We thank the Astrid Fagræus laboratory for the animal husbandry. We acknowledge Science for Life Laboratory, the National Genomics Infrastructure and Uppmax for providing assistance with sequencing and computational infrastructure. We acknowledge Olga Dethlefsen from National Bioinformatics Infrastructure Sweden (NBIS) for bioinformatic advice.

## Author Contributions

**Conceptualization:** Arvin Chireh, Rikard Grankvist, Abdul Kadir Mukarram, Fabian Arnberg, Johan Lundberg, Carsten O. Daub, Staffan Holmin.

**Data curation:** Arvin Chireh, Rikard Grankvist, Abdul Kadir Mukarram.

**Formal analysis:** Arvin Chireh, Rikard Grankvist, Abdul Kadir Mukarram, Carsten O. Daub.

**Funding acquisition:** Staffan Holmin.

**Investigation:** Arvin Chireh, Rikard Grankvist, Mikael Sandell, Fabian Arnberg, Johan Lundberg, Carsten O. Daub, Staffan Holmin.

**Methodology:** Arvin Chireh, Rikard Grankvist, Mikael Sandell, Abdul Kadir Mukarram, Fabian Arnberg, Johan Lundberg, Carsten O. Daub, Staffan Holmin.

**Project administration:** Arvin Chireh, Rikard Grankvist, Staffan Holmin.

**Resources:** Carsten O. Daub, Staffan Holmin.

**Software:** Arvin Chireh, Rikard Grankvist, Abdul Kadir Mukarram.

**Supervision:** Fabian Arnberg, Johan Lundberg, Carsten O. Daub, Staffan Holmin.

**Validation:** Arvin Chireh, Rikard Grankvist, Carsten O. Daub.

**Visualization:** Arvin Chireh, Rikard Grankvist, Abdul Kadir Mukarram.

**Writing – original draft:** Arvin Chireh, Rikard Grankvist.

**Writing – review & editing:** Mikael Sandell, Abdul Kadir Mukarram, Fabian Arnberg, Johan Lundberg, Carsten O. Daub, Staffan Holmin.

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
