## [Decision Letter · Decision Letter 0]

4 Mar 2021

PONE-D-21-02936

Micro-biopsy detects gene expression changes in ischemic swine myocardium

PLOS ONE

Dear Dr. Holmin,

Thank you for submitting your manuscript to PLOS ONE. After careful consideration, we feel that it has merit but does not fully meet PLOS ONE’s publication criteria as it currently stands. Therefore, we invite you to submit a revised version of the manuscript that addresses the points raised during the review process.

 All issues raised by editor and expert reviewers are required.

We look forward to receiving your revised manuscript.

Kind regards,

Vincenzo Lionetti, M.D., PhD

Academic Editor

PLOS ONE

Additional Editor Comments:

Major issues:

1) The authors should add information regarding estimated proportions of cardiomyocyte and endothelial cells.

2) Previous studies have analysed changes of myocardial proteomic profile in infarcted swines following regenerative treatment (doi: 10.1016/j.bbagen.2017.02.006.). The authors should discuss the relevance of their results compared to proteomic analysis in order to detect early hallmarks of cardiac remodeling.

3) Since it is a methodological report, the authors should mention and discuss the following unmentioned articles (Int J Cardiovasc Imaging. 2018 Dec;34(12):1917-1926; PLoS One. 2016 Aug 1;11(8):e0160110) in light of the present data.

Journal Requirements:

"AC, RG, MS, and SH are board members of Microcardix AB, which holds pending patents for the micro-EMB device.

RG, JL, and SH own shares in SmartCella Holding AB, whose subsidiaries have interests in catheter technology and regenerative cardiology."

3. We note that you have a patent relating to material pertinent to this article. Please provide an amended statement of Competing Interests to declare this patent (with details including name and number), along with any other relevant declarations relating to employment, consultancy, patents, products in development or modified products etc. Please confirm that this does not alter your adherence to all PLOS ONE policies on sharing data and materials, as detailed online in our guide for authors http://journals.plos.org/plosone/s/competing-interests by including the following statement: "This does not alter our adherence to  PLOS ONE policies on sharing data and materials.” If there are restrictions on sharing of data and/or materials, please state these. Please note that we cannot proceed with consideration of your article until this information has been declared.

Reviewers' comments:

Reviewer's Responses to Questions

**Comments to the Author**

1. Is the manuscript technically sound, and do the data support the conclusions?

Reviewer #1: Yes

Reviewer #2: Partly

2. Has the statistical analysis been performed appropriately and rigorously? 

Reviewer #1: Yes

Reviewer #2: N/A

3. Have the authors made all data underlying the findings in their manuscript fully available?

Reviewer #1: No

Reviewer #2: Yes

4. Is the manuscript presented in an intelligible fashion and written in standard English?

Reviewer #1: Yes

Reviewer #2: Yes

5. Review Comments to the Author

Reviewer #1: The manuscript “Micro-biopsy detects gene expression changes in ischemic swine myocardium” by Arvin Chireh et al., investigated the ability to detect disease related gene expression changes using micro-EMB. This manuscript is well designed and organized. The focus of this manuscript is new and original and it could have a clinical impact in the future, even if a lot of work has to be done to test the ability of micro-EMB to be used for discover pathophysiological mechanisms in animal models or clinical diagnosis in humans. However, the number of the study samples is very very low. I would suggest to the authors to modified the title in “Micro-biopsy detects gene expression changes in ischemic swine myocardium: a pilot study”. Moreover, I would suggest to describe the work as “a pilot study” through the whole manuscript.

Reviewer #2: Thank you to the authors of the manuscript entitled” Micro-biopsy detects gene expression changes in ischemic swine myocardium” that have tried to evaluate the ability to capture gene expression changes in a myocardial disease model, using a micro-EMB device. The feasibility of Micro-biopsy method of detecting gene expression changes is attractive. However, there are a few issues that need to be addressed.

Overall, it is a technical study with no mechanistic information. It is appreciable that the manuscript provides:

• Highlight the novelty of the study.

• Comment on the number of sample (table 1) is significantly higher than the number of the samples that can

be obtained by other methods.

• Provide the hemodynamic parameters that was measured during the occlusion.

• Please clarify that the undetectable gene expression changes at the later time of the infarction was not due to

the dramatic reduction in tissue specimens.

• Please provide more details about the subsequent micro-EMB procedures

• If there is any histology data for the infarcted tissue.

• Please note that there are several grammatical and typo errors throughout the paper and several sentences

have been structured incorrectly.

6. PLOS authors have the option to publish the peer review history of their article (what does this mean?). If published, this will include your full peer review and any attached files.

Reviewer #1: No

Reviewer #2: No

---

## [Author Response · Author response to Decision Letter 0]

30 Mar 2021

Response to reviewer comments

Response to Editor Comments

1) The authors should add information regarding estimated proportions of cardiomyocyte and endothelial cells.

We did not initially perform this analysis as part of the deconvolution analysis, since our main question of interest was whether there was signs of inflammation and fibrosis at later timepoints. We have now supplemented the analysis with estimations of endothelial cell and cardiomyocyte proportions. As expected, the proportion of cardiomyocytes is lower in the “Day 2” group, supposedly due to a higher proportion of inflammatory cells such as fibroblasts. Please note that Fig 4 is adjusted by fixing the scales to make it easier to understand the absolute changes in cell proportions. Also, significance bars between “day 2” and “day 14” have been added for clarity.

2) Previous studies have analysed changes of myocardial proteomic profile in infarcted swines following regenerative treatment (doi: 10.1016/j.bbagen.2017.02.006.). The authors should discuss the relevance of their results compared to proteomic analysis in order to detect early hallmarks of cardiac remodeling.

Proteomics is an interesting and promising field of research. However, due to the increased variability in proteins, compared to the transcriptome, a larger material is needed to make statistical associations. The micro-EMB tissue sample size is very small (approximately 10 µg), which is probably on lower end of what is technically feasible with current proteomic methods. Since transcriptomic analysis of very small samples is more standardized at the moment, and also more explored in the context of clinical applications, we have chosen to focus on transcriptomic analysis. We have cited this work in the introduction and this is now discussed in the Discussion section (lines 247-252).

3) Since it is a methodological report, the authors should mention and discuss the following unmentioned articles (Int J Cardiovasc Imaging. 2018 Dec;34(12):1917-1926; PLoS One. 2016 Aug 1;11(8):e0160110) in light of the present data.

MRI-guidance is an exciting technique that we are interested in testing together with micro-EMB. We have added a sentence regarding guidance (with appropriate references) to the Discussion (lines 222-225).

Myocardial Expression Analysis of Osteopontin is also an interesting application of transcriptomic analysis of myocardial expression. We have added this citation to the examples of relevant applications of transcriptomics in cardiology.

The data repository information is provided in the current data availability statement (NCBI's Gene Expression Omnibus, GEO Series accession number GSE164962, https://www.ncbi.nlm.nih.gov/geo/query/acc.cgi?acc=GSE164962). As soon as the manuscript is accepted, GEO can be notified to promptly release the data, in accordance to their guidelines. This is now clarified in the statement in the cover letter.

Responses to Reviewer #1: 

"The manuscript “Micro-biopsy detects gene expression changes in ischemic swine myocardium” by Arvin Chireh et al., investigated the ability to detect disease related gene expression changes using micro-EMB. This manuscript is well designed and organized. The focus of this manuscript is new and original and it could have a clinical impact in the future, even if a lot of work has to be done to test the ability of micro-EMB to be used for discover pathophysiological mechanisms in animal models or clinical diagnosis in humans. "

Thank you for reviewing our manuscript! We greatly appreciate your comments and interest in this study.

"However, the number of the study samples is very very low. I would suggest to the authors to modified the title in “Micro-biopsy detects gene expression changes in ischemic swine myocardium: a pilot study”. Moreover, I would suggest to describe the work as “a pilot study” through the whole manuscript."

This is certainly a valid comment; we have edited the manuscript title and main text accordingly.

Responses to Reviewer #2: 

"Thank you to the authors of the manuscript entitled” Micro-biopsy detects gene expression changes in ischemic swine myocardium” that have tried to evaluate the ability to capture gene expression changes in a myocardial disease model, using a micro-EMB device. The feasibility of Micro-biopsy method of detecting gene expression changes is attractive. However, there are a few issues that need to be addressed."

Thank you for reviewing our manuscript!

"Overall, it is a technical study with no mechanistic information. It is appreciable that the manuscript provides:

• Highlight the novelty of the study."

This is the first (pilot) study showing that the novel micro-biopsy technique can be used to detect molecular tissue changes in a disease model. This is partially highlighted in the introduction and discussion parts of the manuscript. We have now clarified in the “Introduction” that the present study shows this for the first time (line 72).

• "Comment on the number of sample (table 1) is significantly higher than the number of the samples that can be obtained by other methods."

It is true that the number of collected biopsies is higher than what is usual in a clinical context of endomyocardial biopsies (with the conventional method). This strategy was selected to prioritize statistical power for this pilot study, rather than optimizing the biopsy protocol for a clinical procedure. This has now been clarified in the “Methods” (lines 271-272) as well as “Discussion” sections (lines 245-247). Nevertheless, the improved safety of micro-EMB probably allows a higher number of biopsies compared to conventional EMB, although this was not evaluated in the present study.

• "Provide the hemodynamic parameters that was measured during the occlusion.

Hemodynamic parameters have been added to the first paragraph of the “Results” section (lines 77-79), and a clarification to the “Study design” paragraph of the “Materials and Methods” (lines 265-266). The data has also been added as a supplemental file (S1 File).

• "Please clarify that the undetectable gene expression changes at the later time of the infarction was not due to the dramatic reduction in tissue specimens."

It is true that the number of sample replicates was few in some of the groups, which could partly explain the failure of detecting later tissue changes. This is now explicitly stated in the Discussion (line 207). 

It is not known whether conventional EMB (with larger tissue specimens) would require fewer replicates to detect late changes compared to micro-EMB, as this has not been studied. Our first study of the micro-biopsy technique (doi: 10.1038/s41598-020-64900-w) showed comparable gene expression patterns between micro-biopsy and conventional EMB samples, however that study did not include any disease model. Including conventional EMBs in the present study would have been hazardous due to a higher risk of LV perforation. Future studies should test the diagnostic capability of micro-biopsies compared to other conventional methods. This has now been clarified in “Discussion” (line 213-216 and 256).

• "Please provide more details about the subsequent micro-EMB procedures"

This has now been clarified in the “Methods” section (lines 322-324).

• "If there is any histology data for the infarcted tissue."

As the presence of infarction was apparent on gross visual inspection at necropsy, and tissue injury was confirmed with cardiac troponin assays, we did not proceed with histology to confirm the infarction. 

Regarding histology of the micro-EMB samples, the samples acquired with micro-EMB are too small to effectively section and analyze with histology. This has been discussed in our previous work (doi: 10.1038/s41598-020-64900-w) and we have now also clarified these circumstances in the Methods section (lines 274-275).

• "Please note that there are several grammatical and typo errors throughout the paper and several sentences have been structured incorrectly."

We have carefully copyedited the manuscript and edited a number of sentences for clarity, please see the manuscript version with tracked changes for complete details.

---

## [Decision Letter · Decision Letter 1]

12 Apr 2021

Micro-biopsy for detection of gene expression changes in ischemic swine myocardium: A pilot study

PONE-D-21-02936R1

Dear Dr. Holmin,

We’re pleased to inform you that your manuscript has been judged scientifically suitable for publication and will be formally accepted for publication once it meets all outstanding technical requirements.

Kind regards,

Vincenzo Lionetti, M.D., PhD

Academic Editor

PLOS ONE

Additional Editor Comments (optional):

Reviewers' comments:

Reviewer's Responses to Questions

**Comments to the Author**

1. If the authors have adequately addressed your comments raised in a previous round of review and you feel that this manuscript is now acceptable for publication, you may indicate that here to bypass the “Comments to the Author” section, enter your conflict of interest statement in the “Confidential to Editor” section, and submit your "Accept" recommendation.

Reviewer #1: All comments have been addressed

Reviewer #2: All comments have been addressed

2. Is the manuscript technically sound, and do the data support the conclusions?

Reviewer #1: Yes

Reviewer #2: (No Response)

3. Has the statistical analysis been performed appropriately and rigorously? 

Reviewer #1: Yes

Reviewer #2: (No Response)

4. Have the authors made all data underlying the findings in their manuscript fully available?

Reviewer #1: No

Reviewer #2: (No Response)

5. Is the manuscript presented in an intelligible fashion and written in standard English?

Reviewer #1: Yes

Reviewer #2: (No Response)

6. Review Comments to the Author

Reviewer #1: (No Response)

Reviewer #2: (No Response)

7. PLOS authors have the option to publish the peer review history of their article (what does this mean?). If published, this will include your full peer review and any attached files.

Reviewer #1: No

Reviewer #2: No

---

## [Editor Report · Acceptance letter]

16 Apr 2021

PONE-D-21-02936R1 

Micro-biopsy for detection of gene expression changes in ischemic swine myocardium: A pilot study 

Dear Dr. Holmin:

I'm pleased to inform you that your manuscript has been deemed suitable for publication in PLOS ONE. Congratulations! Your manuscript is now with our production department. 

Kind regards, 

on behalf of

Prof. Vincenzo Lionetti 

Academic Editor

PLOS ONE